# Organic Compounds Responsible for the Fouling of Ultrafiltration Membrane Treating Algae-Laden Water

**DOI:** 10.3390/membranes13090787

**Published:** 2023-09-12

**Authors:** Edwin Castilla-Rodriguez, Hongde Zhou

**Affiliations:** School of Engineering, University of Guelph, Guelph, ON N1G 2W1, Canada; hzhou@uoguelph.ca

**Keywords:** ultrafiltration, membrane fouling, algal organic matter, biopolymers, soluble microbial products, polysaccharides, proteins

## Abstract

Fouling comparisons of the organic fractions in surface and algae-laden waters make it possible to determine the main compounds responsible for the fouling of ultrafiltration (UF) membranes. This study examined the fouling of UF membranes and its relationship to the characteristics of the organic fractions found in drinking-water supply. Four types of water were prepared by combining natural organic matter (NOM) from lake water with algal organic matter (AOM) from four algae species commonly found in freshwater. Liquid chromatography–organic carbon detection (LC–OCD) and a fluorescence excitation–emission matrix (FEEM) were used to analyze the feed water and permeate to assess the interactions between and fouling behavior of the organic fractions. The results showed that the interaction of large-molecular-weight AOMs on the membrane surfaces and their transport through the membrane pores were the main fouling mechanisms. Polysaccharides followed by protein-like substances were the organic compounds responsible for the fouling of the UF membranes. The fouling affinity of these substances was attributed to two processes, the adsorption of their carboxyl, hydroxyl and cationic groups on the membrane surfaces, and the molecular complexation of their organic groups. The humic substances’ retention was marginal and attributed to the synergetic effects of the polysaccharides and proteins.

## 1. Introduction

The excessive fouling caused by NOM in surface-water treatments persists as a significant operational problem for membrane-filtration systems. This is further compounded by the variations in the composition and properties of NOM, resulting in different extents of fouling. In early studies, humic substances were identified as major foulants of UF membranes in the treatment of surface water [1,2], possibly because they are the most predominant fractions in surface water. However, recent studies showed that NOM can be divided into hydrophobic and hydrophilic fractions because they play different roles in membrane fouling; hydrophobic organic matter is mainly composed of humic substances and fulvic acids from aromatic carbon and carboxyl groups, and hydrophilic organic matter is composed of polysaccharides and proteins from aliphatic carbons and hydroxyl groups [3,4].

Recently, NOM has been characterized using LC–OCD and FEEM to differentiate NOM in groups with similar molecular weights and properties [5,6]. The use of LC-OCD can separate NOM in fractions of different molecular weights and functional groups, including biopolymers, humic substances, building blocks, low molecular weight (LMW) acids, and neutrals [7]. A biopolymer quantified by LC–OCD showed a strong correlation with the fouling of UF-membrane-treated surface water [8]. On the other hand, FEEM spectroscopy is used to identify the presence of florescent organic matter by categorizing its peaks in the contour matrix of excitation and emission wavelengths, including aromatic protein-like, humic substance-like soluble microbial product (SMP)-like, and humic acid-like substances [3,9,10,11]. The aromatic-protein-like substances have been identified as the most significant fraction in terms of causing the fouling of UF membranes [12].

While most of the fouling studies of UF membranes used to treat surface waters have been conducted on water sources with similar organic compositions [13,14], little information is available to compare the role of each organic fraction in the fouling of membranes used to treat surface waters with different organic compositions specifically by comparing the fouling potential of each organic fraction found in surface waters before and during algae blooms. The significance of these organic fractions and their interactions in long-term membrane fouling is still not understood.

A series of experiments were conducted using different concentrations of AOM harvested from algae cultures mixed with NOM from lake water. The algae species included Cyanobacteria, Bacillariophyta, and Chlorophyta because they have been identified as the most predominant groups during algae blooming in surface water [15]. Specially, *Microcystis aeruginosa* and *Merismopedia* sp. were selected as the representative species for the Cyanobacteria group, *Cyclotella* sp. was selected for the Bacillariophyta group, and *Chlorella* sp., was selected for the Chlorophyta group. *Microcystis* blooming is one of the most harmful occurrences in fresh waters [16] and has been increasingly reported worldwide [17].

The objective of this study was to evaluate the UF-membrane fouling of each organic fraction from different AOM/NOM mixtures. The LC–OCD and FEEM techniques were used to characterize the AOM and NOM in the fractions, which helped us to identify the organic compounds responsible for the fouling of UF membranes.

## 2. Materials and Methods

### 2.1. Algal Cultivation

Four predominant algae species were grown in batch cultures at light intensities and nutrient concentrations recommended by the Canadian Phycological Culture Centre (CPCC) at the University of Waterloo. *Chlorella* sp. (CPCC 522) was grown in sterilized bold basal medium adjusted to a pH of 6.8 and with a continuous light intensity of 60–70 µE m^−2^ s^−1^. *Microcystis aeruginosa* (CPCC 633) was grown in sterilized modified bold basal medium with triple amounts of nitrogen and a continuous light intensity of 30–40 µE m^−2^ s^−1^. *Merismopedia* sp. (CPCC 711) was grown in sterilized BG-11 medium adjusted to a pH of 7.5 and a continuous light intensity of 30–40 µE m^−2^ s^−1^. The diatom, *Cyclotella* sp. (CPCC 432), was grown in sterilized CHU-10 medium and 12/12 h light/dark periods at a light intensity of 20–30 µE m^−2^ s^−1^. All cultures were gently aerated to keep them continuously mixed and provide carbon dioxide. The algal cultures were grown in continuous batch reactors, where 5–8% of the algal culture volume was taken and replenished with medium twice a week to simulate actual growth conditions during blooming. Algal growth was monitored during the exponential and stationary phases by counting the cells using Neubauer chamber crystal slide and a light microscope (AmScope—40×–1000× LED binocular), as well as by measuring dry weight and total and dissolved organic carbon.

### 2.2. Test Water

The NOM from Lake Ontario was combined with AOM to make the four test waters used in this study. Water from Lake Ontario was taken from the Burlington Bay Canal, Burlington, Ontario. The AOM was extracted from each algal specie described in Section 2.1. Algae stock samples were centrifuged to separate suspended solids from dissolved organic matter. Equal concentrations of dissolved organic matter per algal specie were added to provide the AOM concentration needed to prepare the surface-water tests. The surface water were prepared with similar characteristics in terms of DOC, pH, hardness, alkalinity, and total suspended solids (TSS), as shown in Table 1. Each test water was prepared with a DOC concentration of 5.0 ± 0.2 mg L^−1^ and with different organic-matter-fraction distributions. Lake (L) was Lake Ontario water with 5.0 mg L^−1^ of NOM as DOC, Lake-Algae (L-A) was Lake Ontario water with 3.3 mg L^−1^ of NOM plus a mixed algae stock, with 1.7 mg L^−1^ of AOM as DOC, Algae-Lake (A-L) was Lake Ontario water with 1.7 mg L^−1^ of NOM plus a mixed algae stock, with 3.3 mg L^−1^ of AOM as DOC, and Algae (A) was milli Q^®^ water with 5.0 mg L^−1^ of AOM as DOC.

### 2.3. UF Testing System

The membrane fouling was investigated under operational conditions relevant to the drinking-water industry. This study was conducted on a bench-scale UF-membrane system built using Suez’s ZeeWeed 1000 PVDF hollow fiber with a nominal pore size of 0.02 µm (molecular-weight cut-off 400 kDa) and outside-in flow path, which were provided by Suez Water Technologies & Solutions (Oakville, ON, Canada). The characteristics of the UF-membrane system are specified in Table 2 and shown in Figure 1.

The UF-membrane systems were operated in dead-end mode, without tank drains, with cycles comprised of 40 min of permeation followed by 30 s of backwash with aeration. The systems were operated, based on manufacturer’s recommendations for surface waters with similar water quality, at a permeate flux of 35 L m^−2^ h^−1^ (LMH), followed by a backwash flux (1.5 times the permeate flux) of approximately 52 LMH, with an aeration intensity of 0.5 m^3^ m^−2^ h^−1^ (0.833 L min^−1^).

### 2.4. UF-Membrane-Fouling Evaluation

The UF-membrane systems were operated under conditions relevant to commercial systems. Commercial UF-membrane systems used to treat similar surface waters are normally operated with partial or full membrane-tank drains to produce system recoveries greater than 90%. Under these conditions, the commercial UF systems reach a solid balance, hours after their initial startup, where the solids accumulated in each filtration cycle equal the solids drained at the end of the cycle. Based on a system recovery of 90%, operational conditions described in Section 2.3 and water quality described in Table 1, it was calculated that a commercial UF system reaches a final solids balance (influent TSS mass is approximately equal to TSS mass drained) at a TSS concentration of approximately 240 mg L^−1^ in the membrane tank.

The UF-membrane systems were operated in dead-end mode without membrane-tank drains in the first 6 filtration cycles to accumulate solids in the membrane tank to approximately 240 mg L^−1^. This helped evaluate the initial membrane fouling. Once the final TSS concentration was achieved, the system was placed in recirculation mode, in which permeate was returned to the membrane tank. This helped evaluate the long-term membrane fouling at steady conditions without additional variables.

Membrane fouling was monitored by measuring the increase in trans-membrane pressure (TMP). Prior to each filtration run, membrane resistance was determined by plotting measured TMP and fluxes while permeating pure water in recirculation mode at different pump speeds. Permeate flow was determined from the weight difference in the collected permeate over two minutes. The speed of the permeate pump was adjusted as needed during the filtration cycles to maintain a constant permeate flow. During each filtration run, raw water was filtered in the UF-membrane system for a week, while the pressure was automatically recorded by the data-acquisition system. The system was operated below the maximum operating TMP (9 psi = 0.62 bar) recommended by the manufacturer. The fluxes and TMP values obtained in each filtration run were used to determine the fouling TMP based on the following equation:
(1)$${\mathrm{TMP}}_{f}=\mathrm{TMP}-{\;\mathrm{TMP}}_{m}$$
where TMP_f_ is the fouling trans-membrane pressure caused by membrane foulant (Pa), TMP is the total trans-membrane pressure (Pa), and TMP_m_ is the trans-membrane pressure of ultrapure water (Pa).

Because the viscosity of water increases with decreasing temperatures, TMP values were temperature-corrected to 20 °C to discard the effect of temperature variations in the membrane fouling. Since all TMP values were adjusted to 20 °C, any change in TMP was attributable to fouling. The TMP values were normalized to 20 °C using the following equation [18]:
(2)TMP20=TMPT×(µ20µT)
where TMP_20_ is the trans-membrane pressure at 20 °C (Pa), TMP_T_ is the trans-membrane pressure at temperature T °C (Pa), µ_20_ is the dynamic viscosity of water at 20 °C (Pa s), and µ_T_ is the dynamic viscosity of water at temperature T °C (Pa s). Prior to each filtration run, the membrane system was operated at 35 LMH in pure water to determine the membrane permeability in clean water. The clean-water permeability adjusted to 20 °C was approximately 425 LMH bar^−1^. The membranes were chemically cleaned after each filtration run and cleaning efficiency was confirmed by a clean-water-permeability test.

### 2.5. Characterization of Organic Matter

Analytical techniques used to characterize AOM and membrane fouling included LC–OCD, FEEM spectrometry, and organic carbon analyzer. Raw water samples were filtered through a 0.45 µm filter in preparation for the analyses. The organic content of the samples was analyzed with a TOC analyzer (TOC-5000; Shimadzu, Japan). The UV-light absorption at 254 nm (UVA_254_) and UV-Vis spectra were measured using a Shimadzu UV-1601 variable-wavelength spectrophotometer and a quartz cuvette with a 1 cm path length.

#### 2.5.1. Liquid Chromatography–Organic Carbon Detector (LC–OCD)

Raw water and permeate samples were analyzed using LC–OCD (Model 8, DOC-Labor, Karlsrube, Germany). The LC–OCD is a form of size-exclusion chromatography combined with organic carbon and nitrogen detectors, and it is used to quantify organic fractions based on their molecular sizes or weights as biopolymers, humic substances, building blocks, low molecular weight (LMW) organic acids and neutrals, and hydrophobic organic carbon [7]. The concentrations of the organic carbon groups previously mentioned were determined based on the integration area under the chromatographic peaks at their corresponding retention times.

The first fractions depicted by the LC–OCD were biopolymers. Biopolymers are hydrophilic fractions comprising polysaccharides and proteins with molecular weights higher than 10 kDa, and non-ionic. These fractions show responses in organic nitrogen detection (OND) due to the presence of nitrogen-containing compounds such as proteins and amino sugars, which are considered hydrophobic [7]. The main compounds in this fraction were polysaccharides, which are the dominant materials in SMP, and extracellular polymeric substances (EPS) from cells. Humic substances were the second type of fraction depicted by the LC–OCD; they consist of humic acid and fulvic acid compounds, mostly hydrophobic, with molecular weights of approximately 1 kDa [7]. Building blocks consist of humic-substance-like materials of lower molecular weight; they are mostly hydrophilic, with molecular weights of approximately 300–500 Da. The LMW acids consist of anionic acids at neutral pH, and are mostly hydrophilic, with molecular weight of less than 350 Da. Finally, LMW neutrals consist of low-molecular-weight and low-ion-density substances. They are mostly hydrophilic alcohols, aldehydes, ketones, sugars, and amino acids, with molecular weights of less than 350 Da [7].

#### 2.5.2. Fluorescence Excitation–Emission Matrix (FEEM) Spectrometry

A Varian Cary Eclipse Fluorescence Spectrophotometer (Palo Alto, CA, USA) was used to measure FEEM of raw water and permeate samples. The FEEM is a 3D fluorescence excitation–emission matrix that measures the fluorescence responses of organic substances in water. The water samples were analyzed to identify protein-like, soluble microbial by-products, as well as humic-substance-like and humic-acid-like substances based on their excitation–emission fluorescence intensities [19]. The spectrophotometer produced emission-intensity values from 280 nm to 550 nm at sequential 5 nm increments of excitation wavelength, from 200 nm to 400 nm. The fluorescence spectra result for a blank using MilliQ water (Millipore, Billerica, MA, USA) was subtracted from the sample results to account for Raleigh scattering effects. The EEM spectrum was quantitatively analyzed by the fluorescence regional integration (FRI) method [19,20].

Three-dimensional EEM fluorescence spectra were analyzed using the FEEM-FRI technique described by Chen et al. [19], to investigate the fluorescence response to the organic substances. The EEM spectra were divided into five regions of organic substances with similar properties based on supporting research by Chen et al. [19], Yao et al. [11], Ou et al. [21], and Yu et al. [12]. Regions I and II included aromatic protein-like substances, such as tyrosine and tryptophan, observed at intensities of excitation wavelength between 200 nm and 250 nm, with emission-wavelength ranges of 280–330 nm and 330–380 nm, respectively. Region III included humic-like substances at intensities of excitation wavelength of 200–250 nm and emission wavelengths f 380–550 nm. Region IV included SMP-like materials at intensities of excitation wavelength greater than 250 nm and emission wavelengths of 280–380 nm. Region V included fulvic-like substances at intensities of excitation wavelength greater than 250 nm and emission wavelengths greater than 380–550 nm.

## 3. Results and Discussion

Algal organic matter—LC-OCD characterization.

The LC–OCD analyses were performed on the AOM extracted from the algae species to compare the compositions of their organic matter. The analyses were conducted on the filtered samples of each of the four algae species and a mix of all the species together. The algae-mix sample consisted of equal AOM masses from each algae species. The percentages of the relative molecular-weight distributions of the AOM fractions for each and the mix of all the algae species are shown in Figure 2. The relative molecular-weight distributions of the AOM fractions were similar for all the algae samples. The organic matter produced by the algae was composed primarily of large-molecular-weight biopolymers–polysaccharides and proteins, followed by humic substances and LMW neutrals. Polysaccharides were the largest AOM fractions, and they had an organic content of approximately 52 ± 3% of the total DOC for all the algae samples. The proteins were the smallest fractions, with organic contents of approximately 9.6 ± 1.7%. The humic substances were the second-largest fractions, with organic contents of approximately 15.4 ± 2.7%, followed by LMW neutrals, with approximately 15 ± 4.2% of the total DOC for all algae samples. The building blocks and LMW acids together comprised less than 10% of the total DOC content.

### 3.1. Algal Organic Matter—FEEM Characterization

Fluorescence regional integration analyses were performed on the EEM fluorescence spectra to quantify the regional volume of the intensities and percentage fluorescence response of the AOM. The relative fluorescence-response distributions of the AOM for each and for the mix of all the algae species are shown in Figure 3. The relative fluorescence-response distributions of the AOM regions were similar for all the algae samples. The fluorescence response of the organic matter produced by the algae was divided into two groups of similar characteristics, SMP-like substances and humic matter (HM)-like substances. The SMP-like substances are a pool of organic substances, from microbial metabolism and algae decay, including polysaccharides, proteins, organic acids, and structural compounds of algae cells, which were represented by the fluorescence responses of regions I, II, and IV. These compounds are the building blocks that make up algae cells and consist of polymeric macromolecules with broad molecular-weight distributions that vary from 1 kDa to 100 kDa [22]. The combined fluorescence responses of the SMP-like substances produced by the algae were approximately 47.8 ± 7.1% of the total volume of the intensities.

Humic-matter-like substances consist of humic acid and fulvic-acid-like substances, and they are represented by the fluorescence responses of regions III and V, respectively. These fractions had the largest fluorescence responses, with percentage distributions of approximately 26 ± 1.8%, and 26.2 ± 3% of the total volume of the intensities for all the algae samples, respectively. Fulvic acids and humic acids are mixtures of complex carboxyl and phenolic acid molecules with aromatic and aliphatic compounds, with average molecular weights of approximately 1 kDa and 3 kDa, respectively [22].

### 3.2. UF-Membrane-Fouling Responses to Various Natural Organic Fractions

The fouling potential of each of the four test waters was evaluated in terms of the TMP change and TSS accumulation per filtration cycle, and the values for a period of 24 h are shown in Figure 4. The waters had similar characteristics with different organic fraction distributions, as described in Section 2.2, which allowed a comparison of their fouling potential. The membrane systems were similarly operated under the conditions described in Section 2.3. The membranes exhibited different fouling behaviors for each test water, whereas the TSS retention was similar, with an accumulation rate of approximately 38 mg L^−1^ per cycle in the first cycles. As illustrated in Figure 4, a higher level of fouling TMP was observed in the membranes treating the waters with higher AOM contents. The algae water showed the highest level of fouling TMP, with a delta fouling TMP (difference between before backwash (BB) and after backwash (AB) TMP)that increased from 0.10 bar in the first cycle to 0.32 bar in the sixth cycle. The TMP increase in the first six cycles (4 h) resulted in an overall slope of approximately 0.11 bar h^−1^. The suspended solids of this water type consisted of algae cells. During the filtration, the algae cells were size-excluded and accumulated on the membrane surface, leading to the formation of compact cake/gel layers, which resulted in TMP increases. This membrane-fouling mechanism was considered a major factor in membrane fouling [3,23]. However, the high content of AOM induced further TMP increases, probably due to its interaction with the algae cells and membrane surface.

The A-L and L-A waters showed TMP increases that resulted in slopes of approximately 0.095 and 0.050 bar h^−1^, respectively. The rapid increase in fouling TMP in these waters can be attributed to the membrane retention and absorption of larger-molecular-weight organic matter, followed by their densification and increases in bulk viscosity (Figure 4). The steep increase in TMP per cycle was accompanied by high rates of biopolymer retention [24] and compaction on the membrane surface [25], presumably due to the biopolymers’ adsorption on the pore walls, leading to irreversible fouling [13]. After the six initial filtration cycles, the delta TMP gradually decreased as the solids resettled in the membrane tank. The suspended solids of these waters were a mixture of algae cells and solids found in lake water, as described in Table 1. The TMP increase in these two water types was proportional to the algae-cell and AOM contents, resulting in higher TMP increases in the A-L water than in the L-A water.

The membrane fouling treating the Lake water had the least fouling TMP, with a delta TMP that slowly increased from 0.020 bar in the first cycle to 0.050 bar in the sixth cycle. This increase in TMP was attributed only to the solids’ accumulation on the membrane surface. After the first six filtration cycles, the delta TMP gradually decreased back to its initial value for the remainder of the run. The low fouling of the TMP during the treatment of the Lake water may be attributed to the low NOM retention by the membranes. The DOC in the Lake water consisted mostly of humic substances. These substances have much lower molecular weights in comparison to the membrane molecular-weight cut-off and tend to pass through membranes. The suspended solids of this water type consisted of solids found in the lake water, as described in Table 1. The TMP increase in this water type was mild and expected of surface water without AOM content.

### 3.3. Interaction between NOM Fractions and UF Membranes—LC–OCD Characterization

The organic fractions in the four test waters described in Section 2.2 were measured and used to evaluate their fouling propensities during the UF-membrane filtration. The interactions between the organic fractions within the membranes were evaluated by comparing their retentions in the membrane tank over time. The organic fraction retention was the concentration difference between the water in the membrane tank and its corresponding permeate (per). The changes in the organic matter fractions in the first six filtration cycles measured by the LC–OCD for each test water are shown in Figure 5. The DOC retention by the membranes during the filtration increased with the addition of the AOM in the following order, as shown in Figure 5: Lake < Lake-Algae < Algae-Lake < Algae. The membranes treating the Algae and A-L waters retained the most influent DOC, producing the lowest permeability of DOC of all the test waters. The DOC in these waters consisted largely of polysaccharides and proteins, which were mostly retained by the membranes. The humic substances and building-block concentrations decreased with the addition of the AOM. These substances mostly passed through the membranes and were the predominant fraction in the Lake water. The LMW neutrals and acid concentrations varied slightly and were less than 17% of the total DOC for all the test waters. These substances mostly passed through the membranes; however, the LMW neutrals showed slightly higher retention when the Algae water was filtered.

The membranes treating the L-A and Lake waters retained less influent DOC and produced more permeable DOC than the Algae water. The biopolymer–polysaccharides and proteins were mostly retained within the membranes in the first six filtration cycles. Their retention was mainly attributed to size exclusion as a result of their much larger MW (10–10,000 kDa) in comparison to the membranes’ MW cut-off (400 kDa). In addition, the biopolymers acted as attachment agents between the molecules and onto the membrane surfaces, resulting in their high retention and subsequent accumulation, which led to much less permeable DOCs [26]. This interaction may be attributed to the charge density, which facilitates linkage to the membrane surface. Polysaccharides in biopolymers are mostly hydrophilic, with an affinity for PVDF-membrane surfaces as a result of the hydrogen bonding between the fluorine in the PVDF and the oxygen in the hydroxyl group [27].

Hydrophobic and transphilic protein-like substances play an important role in membrane fouling. Proteins and amino sugars are considered mostly hydrophobic [7], with transphilic properties, and tend to react within membranes, aggregating and accumulating within membrane pores. The hydrophobic properties of protein substances facilitated the retention of large-molecular-size hydrophilic polysaccharides, increasing membrane fouling. Similarly, Yu et al. [12] suggested that protein-like substances were the major foulants of UF membranes used to treat water with *M. aeruginosa*. Membrane fouling was attributed to the transphilic adhesion of the protein-like substances on the membrane surfaces.

The humic substances, building blocks, and LMW acids showed no initial retention by the membranes. As shown in Figure 5, their retentions were initially insignificant in comparison to the biopolymer; however, in subsequent permeate cycles, they experienced some retention, which was mostly attributed to the synergetic interactions with the biopolymers. On the other hand, the LMW neutrals showed some affinity with the membrane surface. This fraction showed a moderate initial retention that gradually increased with subsequent cycles. Their initial retention can be attributed to a neutralizing interaction with the proteins, which may have led to a reduction in the charge density of AOM [28]. Their increased retention in the subsequent filtration cycles suggests additional synergistic effects with the proteins and polysaccharides in the bulk liquid.

Specific UV absorbance (SUVA) values in the membrane tank were indicators of the predominant fractions in the waters. The SUVA values for the waters with predominantly humic-substance contents were higher than those for the waters with polysaccharides and proteins. For instance, the SUVA values for the Lake water decreased from 2.95 L m^−1^ mg^−1^ in the first cycle to 1.3 L m^−1^ mg^−1^ in the sixth cycle, whereas, for the Algae water, they decreased from 0.99 L m^−1^ mg^−1^ in the first cycle to 0.17 L m^−1^ mg^−1^ in the sixth cycle. The much lower SUVA values for the Algae water were due to the much higher accumulation of polysaccharides and proteins in the membrane tank. High SUVA values are associated with greater aromatic contents of humic-like substances. The SUVA changes for all the test waters during the filtration can be seen in Figure 5.

### 3.4. NOM Fractions’ Interactions with UF Membranes—FEEM Characterization

We performed FRI analyses on the EEM-fluorescence responses of the organic fractions from each test water and permeate to evaluate their interactions within the membranes. The changes in the organic matter fractions in the first six filtration cycles measured by FEEM–FRI for each test water are shown in Figure 6. The organic fractions for each water type had distinctive fluorescence responses. In all the waters, the humic- and fulvic-like substances emitted more and stronger fluorescence responses than other fractions. Their stronger fluorescence responses were attributed to their aromatic properties, with phenolic and carboxylic substituents [29], which give them an extended range of excitation and emissions. Although the humic- and fulvic-like substances were the larger fractions in all the waters, their retentions were low in comparison to the SMP-like and protein-like substances.

Out of all four test waters, the Lake water demonstrated the highest contents of humic-matter-like substances (humic- and fulvic-like substances). The high content of humic matter in the lake water agrees with the understanding of the NOM composition of surface water. The humic-matter-like substances showed low retention by membranes. Their low retention and interaction within the membranes were attributed to their hydrophobic properties and relatively small molecular sizes compared to the membrane pores. Their accumulation in the bulk liquid and their retention by the membranes in the subsequent filtration cycles slightly increased with the increase in the SMP-like substances (SMP-like and protein-like substances). As a result, their subsequent retention was attributed to the synergetic effects with the SMP-like substances. The Algae water demonstrated the highest levels of SMP-like substances. These substances showed higher levels of retention than the humic-matter-like substances. Their higher levels of retention and interaction within the membranes were mainly attributed to their relatively large molecular sizes compared to the membrane pores. These substances comprise polysaccharides, proteins, and structural compounds of algae cells that can be compared with the biopolymers revealed by the LC–OCD.

### 3.5. Effects of Organic Matter Fractions on UF-Membrane Fouling—LC–OCD Characterization

In order to evaluate the effect of the organic matter fractions’ retention in the UF-membrane fouling, the accumulated organic matter fractions’ concentrations in the membrane tank were plotted against their corresponding fouling TMP in Figure 7. As the figure shows, the fouling TMP increased as the biopolymer concentrations in the membrane tank increased. There was a strong correlation between the biopolymer–polysaccharide and the protein concentrations in the membrane tank with the increases in fouling TMP. These fractions have much larger molecular sizes in comparison to the membrane-molecular-size cut-off, which facilitated their retention and the progressive increase in the fouling-TMP rate. Similarly, Kimura et al. [30] found a significant correlation between biopolymer concentrations and the fouling rates of MF membranes treating surface waters. In their study, the biopolymer concentrations in surface waters were suggested as key indicators of membrane fouling. The humic substances, building blocks, and LMW neutrals and acids showed no correlations with the fouling TMP. Their very low retentions were mostly attributed to synergetic interactions with the biopolymers.

The deposition of algae cells and large-molecular-weight AOM on the membrane surfaces was the main fouling mechanism observed in this study, along with the transport of AOM through membrane pores. The waters with high AOM contents showed rapid increases in fouling TMP, which was attributed to the initial AOM absorption within the membrane pores, specifically by larger MW biopolymers, followed by the densification and increases in bulk viscosity [26]. The biopolymers showed great affinity with the PVDF membrane, acting as attachment agents between molecules and onto the membrane surfaces. Polysaccharides with high MW promote intermolecular association, resulting in poor solubility [31]. The carboxyl, hydroxyl, and cationic groups of their internal structures are free of mutual interactions and can be adsorbed [29]. This affinity can lead to a rapid increase in the bulk density until its viscosity becomes non-Newtonian, resulting in a steep increase in TMP. Similarly, Pongpairoj et al. [32] suggested that AOM absorption and deposition on the membrane surface was the main fouling mechanism. Their research suggested that AOM macromolecules change their conformation and concentration-polarization layer upon entering membrane pores, resulting in increases in the transmission of AOM and in TMP.

The water with high levels of humic substances showed low levels of fouling TMP. Humic substances are nonionic polyelectrolytes with hydrophobic properties and MW rates of approximately 0.5–10 kDa. The initial retention of these substances was insignificant. This was attributed to their LMW and hydrophobic properties; the substances exhibit low affinity with the membrane surface, mostly passing through the pores. At neutral pH, these polyelectrolytes assume extended shapes as a result of intramolecular electrostatic repulsion [29]. However, as the bulk density increased, due to the accumulation of organic matter in the membrane tank, it is believed that the humic substances were adsorbed onto the polysaccharides and proteins. The adsorption may have occurred as a result of the interaction between hydroxyl and carboxyl groups, creating polymeric clusters that were easily retained by the membranes. This indicates that these substances are not fouling fractions of NOM and only play an interactive role with polysaccharides and proteins. Similarly, Tian et al. [8] found no significant correlation between humic-substance content and the fouling of UF membranes treating Lake Tegel and Berlin Canal water. In their research, membrane fouling was mainly correlated with the biopolymer content in the water.

### 3.6. Effects of Organic Matter Fractions on UF-Membrane Fouling—FEEM Characterization

The organic matter fractions that accumulated in the membrane tank were depicted using the FEEM–FRI technique and plotted against their corresponding fouling TMP in Figure 8 to evaluate their fouling effect. As the figure shows, the humic- and fulvic-like substances showed no correlations with the membrane-fouling TMP. The retention levels of these substances were similar, despite the fouling TMP. This suggests that the retention of these substances had no effect on the membrane fouling, indicating that they were not the organic fractions responsible for the membrane fouling. On the other hand, the fouling TMP increased with the increase in the SMP-like and the tyrosine- and tryptophan-protein-like substances in the membrane tank. The high retention of these substances was mainly attributed to their relatively large molecular sizes compared to the membrane pores. This suggests that these substances were mostly responsible for the membrane fouling. Similarly, Zhao et al. [33] found that protein-like substances were more likely to cause UF-membrane fouling during algae-harvesting filtration.

The SMP-like and protein-like substances seem to be related to biopolymers, since that these substances are composed of polysaccharides and protein-like substances from microbial metabolism and algae decay. Similar observations were made by Yu et al. [12] and Henderson et al. [34], who compared the results of both techniques and found that the protein-like substances were highly correlated with the fouling of UF membranes treating algae-laden water and secondary effluent, respectively.

## 4. Conclusions

A higher retention of polysaccharides and protein-like substances was observed on UF membranes treating surface waters with higher AOM contents. The interaction of these substances on the membrane surfaces helped retain lower-molecular-weight organic matter, producing permeates with lower DOC concentrations.The deposition of algae cells and large-molecular-weight AOM on the membranes’ surfaces, along with the transport of AOM through the membranes, was the main fouling mechanism observed in this study.The SMP-like substances and biopolymers were identified as the groups of organic fractions responsible for the fouling of the UF membranes. The fouling TMP increased with the accumulation of these substances in the membrane tank. These substances seem to be related, since biopolymers are composed of polysaccharides and protein-like substances from microbial metabolism and algae decay.The fouling affinity of the polysaccharides and protein-like substances was attributed to two processes: the adsorption of their carboxyl, hydroxyl, and cationic groups on the membrane surfaces and the molecular complexation of their organic groups with multi-valent inorganic ions.The retentions of the humic and fulvic substances were low and similar, despite the fouling TMP. This suggests that their retention had no effect on the membrane fouling and could be attributed to synergetic effects with polysaccharides and protein-like substances.The building blocks and LMW acids showed no significant retention. The LMW neutrals’ retention was low, suggesting that these substances may have interacted with the proteins and polysaccharides.The TMP increase was proportional to the algae-cell and AOM contents of the water evaluated. The high content of AOM induced further TMP increases, probably due to its interaction with the algae cells and the membranes’ surfaces.

## Figures and Tables

**Figure 1 membranes-13-00787-f001:**
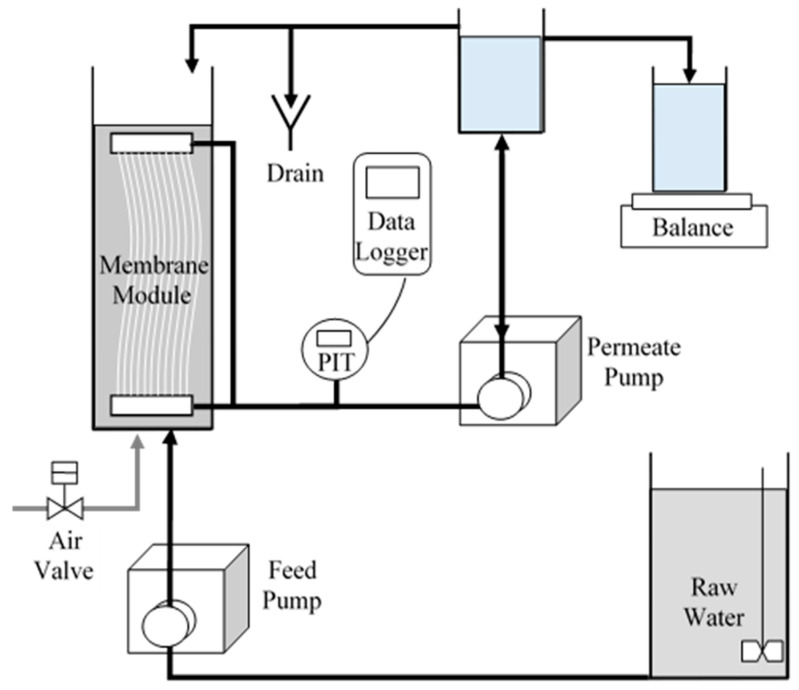
Setup of UF-membrane-testing apparatus.

**Figure 2 membranes-13-00787-f002:**
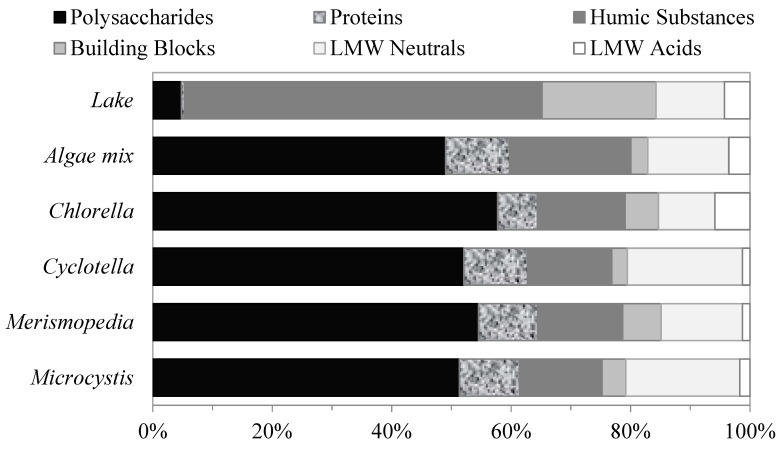
Relative contents of AOM fractions of different algae determined using LC–OCD method.

**Figure 3 membranes-13-00787-f003:**
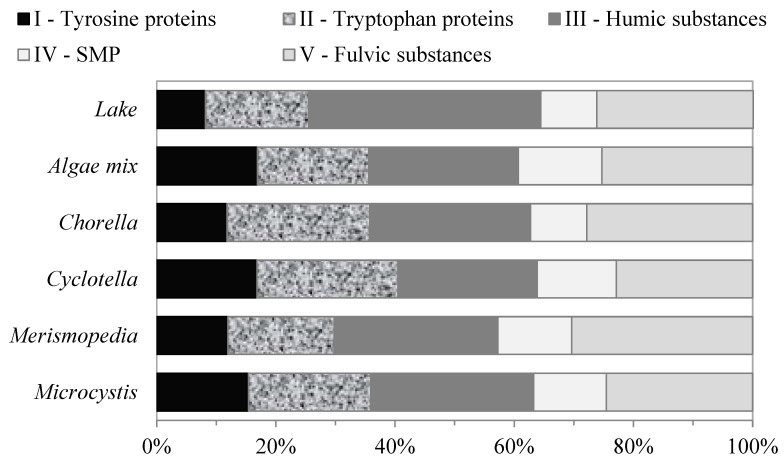
Relative contents of AOM fractions of different algae determined using FEEM–FRI method.

**Figure 4 membranes-13-00787-f004:**
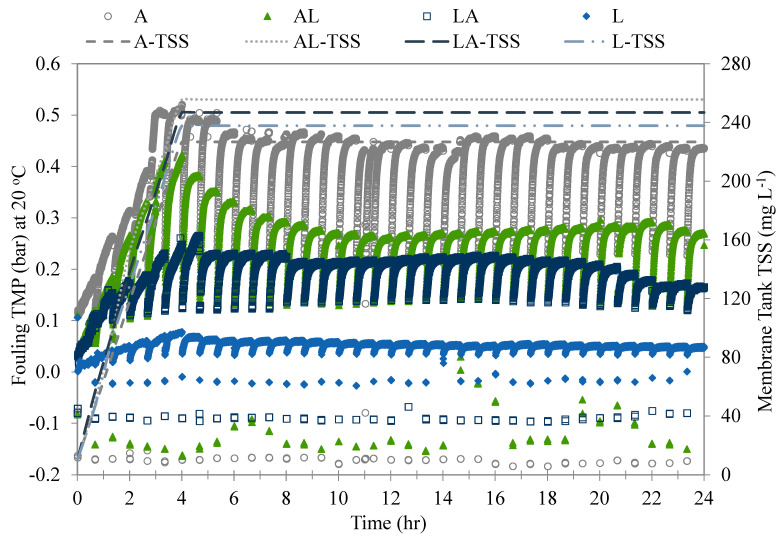
Comparison of fouling behavior and TSS accumulation among four test waters during operation.

**Figure 5 membranes-13-00787-f005:**
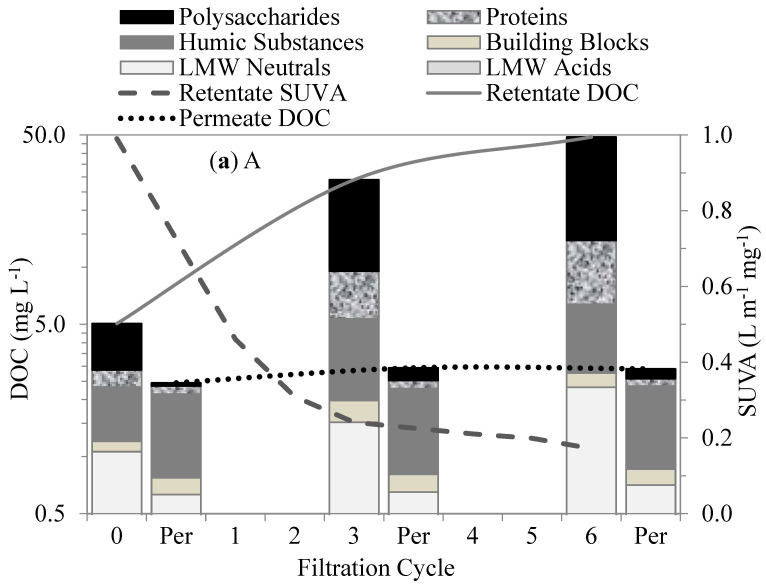
Changes in organic fractions and SUVA after different cycles of membrane filtration using LC–OCD method: (**a**) A, (**b**) AL, (**c**) LA, and (**d**) L.

**Figure 6 membranes-13-00787-f006:**
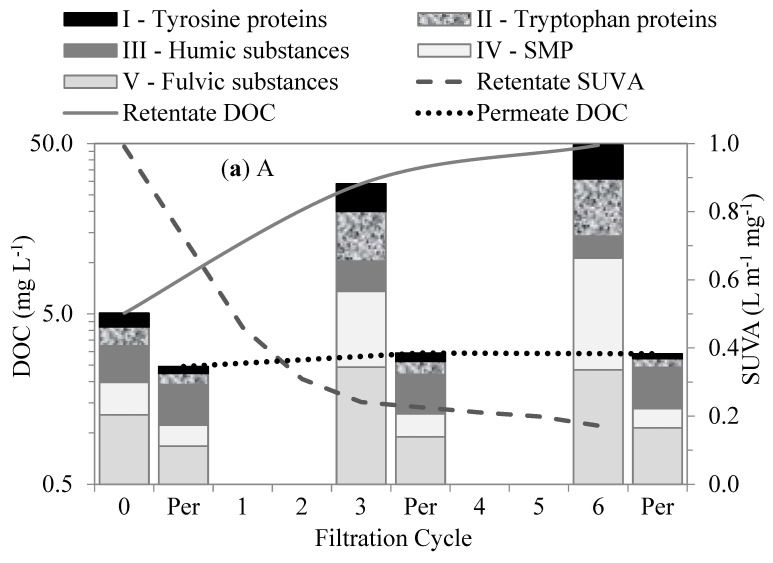
Changes in organic matter fractions and SUVA after different cycles of membrane filtration using FEEM method: (**a**) A, (**b**) AL, (**c**) LA, and (**d**) L.

**Figure 7 membranes-13-00787-f007:**
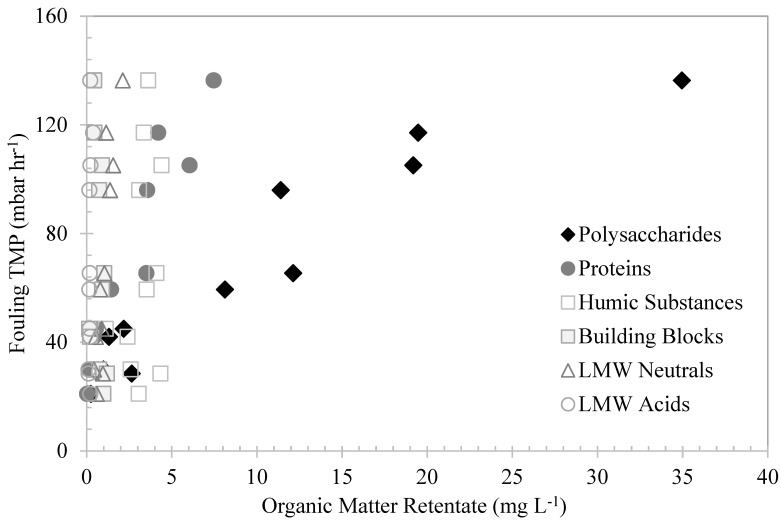
The effects of different organic fractions in the retentates measured by LCD–OCD method on fouling TMP.

**Figure 8 membranes-13-00787-f008:**
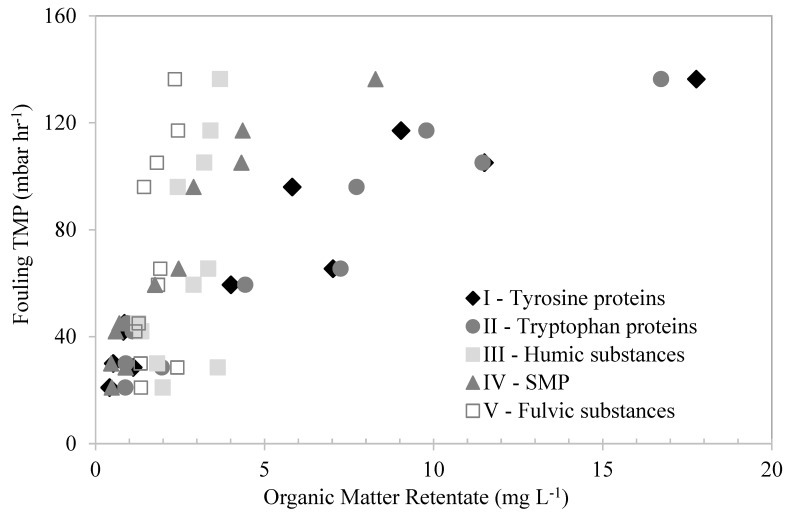
The effects of different organic fractions in the retentates, measured by FEEM–FRI method, on fouling TMP.

**Table 1 membranes-13-00787-t001:** Main characteristics of test waters.

Test Water	Unit	L	LA	AL	A
Lake DOC	mg L^−1^	5.0	3.3	1.7	-
Algal DOC	mg L^−1^	-	1.7	3.3	5.0
UVA	m^−1^	0.150	0.100	0.065	0.050
SUVA	L m^−1^ mg^−1^	2.9	2.0	1.3	1.0
Turbidity	NTU	8	20	27	20
pH		7	7	7	7
Hardness	mg L^−1^ CaCO_3_	120	120	120	120
Alkalinity	mg L^−1^ CaCO_3_	60	70	70	60
Lake TSS	mg L^−1^	10	6.6	3.3	-
Algal TSS	mg L^−1^	-	3.3	6.6	10

**Table 2 membranes-13-00787-t002:** Main specifications of UF-membrane-testing apparatus.

Items	Value or Description
Fiber outside/inside diameter, mm	0.95/0.47
Nominal pore side, µm	0.02
Module-membrane area, m^2^	0.1
Membrane-tank volume, L	0.6
Permeate-tank volume, L	0.3
Feed-tank volume, L	50

## Data Availability

Not applicable.

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
