# Peer review of "Organic Compounds Responsible for the Fouling of Ultrafiltration Membrane Treating Algae-Laden Water"

_membranes, 2023, doi:10.3390/membranes13090787_

Round 1

Reviewer 1 Report

I found that the paper is very similar to a paper that has been published in 2021 (Laksono, S., ElSherbiny, I. M., Huber, S. A., & Panglisch, S. (2021). Fouling scenarios in hollow fiber membranes during mini-plant filtration tests and correlation to microalgae-loaded feed characteristics. Chemical Engineering Journal, 420, 127723.). This paper is indeed included in the paper's reference list, but is not discussed at all in the paper's text (it is not even referenced there)

Minor English editing recommended.

Author Response

Thank you for your comments. The paper described in your comment (Laksono, S., ElSherbiny, I. M., Huber, S. A., & Panglisch, S. (2021). Fouling scenarios in hollow fiber membranes during mini-plant filtration tests and correlation to microalgae-loaded feed characteristics. Chemical Engineering Journal, 420, 127723.) focuses on the membrane fouling rate and mechanisms of four pure water-algae solution using four different algae species than the ones used in our research. There are many differences in that paper in comparison to our research, some of the differences include:

  • The dissolved organic carbon (DOC) concentrations of the testing waters are not realistic to what is found during algae blooms, in China, Canada, Africa, or rest of the world.
  • The DOC of each testing water was different; as a result, that paper fails standardizing the membranes fouling rates with the DOC concentrations. It is understood that the membrane fouling is directly related to the DOC concentration and algae cells content in the water. Lower membrane fouling is expected for membrane systems treating water with lower DOC concentrations and algae cell content.
  • The membrane chemistry used in that research is PES, ours was PVDF. PVDF is the most common membrane used in drinking water treatment in the world due to its properties. As a result, the fouling rates in both researches are different.
  • The paper focused on the type of fouling such as reversable, irreversible, cake, and pore blocking.   
  • The paper was mentioned to compare the steep increase in TMP per cycle as a result of biopolymer accumulation. We did not find the need to mention more about the paper as per the differences described above.

Thanks

Edwin Castilla-Rodriguez

Reviewer 2 Report

Dear Editor

 Manuscript entitled "Organic compounds responsible for the fouling of ultrafiltration membrane treating algae laden water" for consideration for publication in Membranes. This manuscript described the behavior of the fouling of commercial PVDF UF membranes and its relationship to the characteristics of the organic fractions found in drinking water supply. Various techniques were used such as Liquid chromatography-organic carbon detection and fluorescence excitation-emission matrix analytical techniques to analyze the feed water and permeate to assess the interaction and fouling behavior of each organic fractions.

This study is seems good, and the topic is of significant importance. Thus, it is recommended for publication with major revision. I have the following comments for the author to consider in their revised manuscript before the publication in Membranes.

Comments to the Authors:

1.      The abstract has been wordy written and should be enriched by adding the significant results.

2.      There are some typos that should be corrected before publication. For example: “NOM into fractions of different molecular weight”

3.      The introduction has been well written but still there is a vaguely written in the literature review regarding the types of fouling and anti-fouling membrane processes. My suggestion is to add information on the types of fouling and their relation with the types of waters to filling the gaps between the literature and the current work. I suggest to see these updated articles  (https://doi.org/10.3390/membranes12111043) and (https://doi.org/10.30684/etj.2022.134070.1219)

4.      Some information are missing in the setup of the UF membrane apparatus and wrong direction of flow.

5.      Is the permeate pump working as a vacuum pump with TMP of 0.62 bar?

6.      Line 134, “(lmh) followed” “approximately 52 lmh” should be (LMH)

7.      Line 147, “dead-mode without tank” should be “dead-end mode”

8.      The authors should add a reference for Equation 2.

9.      Line 301 “probably due to its interaction with the algae cells and membrane surface.” The authors simply describe their observation, but interpretation of facts the reader is not enough to find. The authors should add details on interaction mechanism between algae cells and surface of the membrane.

10.  The title of section 3.3 “Interaction between NOM fractions with UF membranes – LC-OCD characterization” does not reflect the scientific content of the section.

11.  Line 400, “Their higher fluorescence responses were attributed to their aromatic properties with phenolic and carboxylic substituents that give them an extended range of excitation and emissions.” The authors should add a reference to confirm your explanation!

The English quality its seems good

Author Response

  1. The abstract has been wordy written and should be enriched by adding the significant results.

Response: the research is about finding the organic matter fraction responsible of the fouling of UF membranes treating algae laden water. The main result is included in the abstract “The results showed that the interaction of large molecular weight AOM on the membrane surface along with their transport through the membrane pores were the main fouling mechanisms. Polysaccharides followed by protein-like substances were the organic compounds responsible for the fouling of UF membranes. The fouling affinity of these substances was attributed to two processes, the adsorption of their carboxyl, hydroxyl and cationic groups on the membrane surface and molecular complexation of their organic groups.”

  1. There are some typos that should be corrected before publication. For example: “NOM into fractions of different molecular weight”

Response: corrected.

  1. The introduction has been well written but still there is a vaguely written in the literature review regarding the types of fouling and anti-fouling membrane processes. My suggestion is to add information on the types of fouling and their relation with the types of waters to filling the gaps between the literature and the current work. I suggest to see these updated articles (https://doi.org/10.3390/membranes12111043) and (https://doi.org/10.30684/etj.2022.134070.1219)
  2. Some information are missing in the setup of the UF membrane apparatus and wrong direction of flow.

Response: the micro positive displacement pumps were able to reverse the flow. The pumps were wired and controlled to permeate and backwash in cycles comprised of 40 minutes of permeation followed by 30 seconds of backwash with aeration to simulate the operation of UF membrane full scale systems.

  1. Is the permeate pump working as a vacuum pump with TMP of 0.62 bar?

Response: the pump was operating in negative pressure during permeation and positive pressure during backwash. When the fouling TMP is calculated the resulting fouling TMP pressure is positive during permeation and negative during backwash as shown in Figure 4.

  1. Line 134, “(lmh) followed” “approximately 52 lmh” should be (LMH)

Response: corrected

  1. Line 147, “dead-mode without tank” should be “dead-end mode”

Response: corrected

  1. The authors should add a reference for Equation 2.

Response: included

  1. Line 301 “probably due to its interaction with the algae cells and membrane surface.” The authors simply describe their observation, but interpretation of facts the reader is not enough to find. The authors should add details on interaction mechanism between algae cells and surface of the membrane.
  2. The title of section 3.3 “Interaction between NOM fractions with UF membranes – LC-OCD characterization” does not reflect the scientific content of the section.

Response: Section 3.3 shows the observed interaction of the NOM fractions with UF membranes. The section describes the retention, interaction and passage of the different NOM fractions characterized by the LC-OCD.  

  1. Line 400, “Their higher fluorescence responses were attributed to their aromatic properties with phenolic and carboxylic substituents that give them an extended range of excitation and emissions.” The authors should add a reference to confirm your explanation!

Response: reference to the phenolic and carboxylic substituents composition of the humic substances was added to the line 400.

Reviewer 3 Report

Very good work! Few comments are following:

Figure 2 and lines 244-253: the polysaccharides content in algae mix sample was lower than the corresponding in samples obtained from single cell species from where the mix was prepared; how is it explained?

Lines 285-286: results in Figure 4 correspond to 24 hours of operation; what is expected at longer times?

Figure 4: the TMP patterns in LA and AL samples seem to be closer between the 10th and 18th hours of operation, after which a larger discrepancy was observed; where could it be attributed?

Figures 5 and 6: results are presented for the first 6 cycles of filtration; is it assumed that at longer operation times a similar behavior will be observed?

Author Response

Figure 2 and lines 244-253: the polysaccharides content in algae mix sample was lower than the corresponding in samples obtained from single cell species from where the mix was prepared; how is it explained?

Response: the polysaccharides portion of the algae mix samples was within the observed portion 52±3% of the total DOC for all algae samples. The polysaccharide portion of each algae specie was not a fixed value.

Lines 285-286: results in Figure 4 correspond to 24 hours of operation; what is expected at longer times?

Response: In this research, it was observed that the TMP after 24 hours of operation remained similar to the values observed in the last 12 hours of operation. The UF membrane systems were operated in dead-end mode without tank drains in the first 6 filtration cycles to accumulate solids in the membrane tank to approximately 240 mg L-1. This helped evaluate the initial membrane fouling. Once the final TSS concentration was achieved, the system was placed into recirculation mode where permeate was returned to the membrane tank. This helped evaluate the long-term membrane fouling at steady conditions without additional variables. The TMP observed after the sixth cycle remained similar for the remaining of the 24-hours run.

Figure 4: the TMP patterns in LA and AL samples seem to be closer between the 10th and 18th hours of operation, after which a larger discrepancy was observed; where could it be attributed?

Response: after the sixth cycle of operation, the system was placed into recirculation mode where permeate was returned to the membrane tank. The variation on TMP after the sixth cycle corresponded to the repositioning of the solids in the aeration tank after backwash with aeration. The solid concentration in the aeration tank remained constant after the sixth cycle.

Figures 5 and 6: results are presented for the first 6 cycles of filtration; is it assumed that at longer operation times a similar behavior will be observed?

Response: the dead-end operation of the UF membrane system for six cycles without membrane tank drains is equivalent to the operation of a commercial UF membrane system with partial membrane tank drains at a typical recovery of 90%. Both systems treating the same water quality described in the Section 2.3 of this research. The membrane fouling in the first six cycles is what a new commercial UF membrane system will experience for more than a day until reaches a final solids balance (influent TSS mass is approximately equal to TSS mass drained) at TSS concentration of approximately 240 mg L-1 in the membrane tank. Once the UF membrane system reaches that balance, it is expected that the membrane fouling plateau with marginal increase over time.

Round 2

Reviewer 1 Report

No further comments. Could be published in the present form

Reviewer 2 Report

Dear Editor

I would like to inform you that the authors answer all of the comments, therefore, I recommend to accept it for publication in Membranes

Regards